# Extracellular Vesicles from Steatotic Hepatocytes Provoke Pro-Fibrotic Responses in Cultured Stellate Cells

**DOI:** 10.3390/biom12050698

**Published:** 2022-05-13

**Authors:** Maria Teresa Koenen, Elisa Fabiana Brandt, Dawid Marcin Kaczor, Tim Caspers, Alexandra Catharina Anna Heinzmann, Petra Fischer, Daniel Heinrichs, Theresa Hildegard Wirtz, Christian Trautwein, Rory R Koenen, Marie-Luise Berres

**Affiliations:** 1Medical Department III, Faculty of Medicine, University Hospital RWTH Aachen, 52074 Aachen, Germany; teresa.koenen@rheinmaasklinikum.de (M.T.K.); elisa.brandt@rwth-aachen.de (E.F.B.); tcaspers@ukaachen.de (T.C.); pfischer@ukaachen.de (P.F.); d.heinrichs@fz-juelich.de (D.H.); thwirtz@ukaachen.de (T.H.W.); ctrautwein@ukaachen.de (C.T.); 2CARIM—Department of Biochemistry, School for Cardiovascular Sciences, Maastricht University, 6229ER Maastricht, The Netherlands; d.kaczor@maastrichtuniversity.nl (D.M.K.); alexandra_buskens@yahoo.nl (A.C.A.H.)

**Keywords:** stellate cell, liver fibrosis, extracellular vesicles, non-alcoholic fatty liver disease, metabolic-associated fatty liver disease

## Abstract

Hepatic steatosis and chronic hepatocyte damage ultimately lead to liver fibrosis. Key pathophysiological steps are the activation and transdifferentiation of hepatic stellate cells. We assessed the interplay between hepatocytes and hepatic stellate cells under normal and steatotic conditions. We hypothesized that hepatocyte-derived extracellular vesicles (EVs) modify the phenotype of stellate cells. By high speed centrifugation, EVs were isolated from conditioned media of the hepatocellular carcinoma cell line HepG2 under baseline conditions (C-EVs) or after induction of steatosis by linoleic and oleic acids for 24 h (FA-EVs). Migration of the human stellate cell line TWNT4 and of primary human stellate cells towards the respective EVs and sera of MAFLD patients were investigated using Boyden chambers. Phenotype alterations after incubation with EVs were determined by qRT-PCR, Western blotting and immunofluorescence staining. HepG2 cells released more EVs after treatment with fatty acids. Chemotactic migration of TWNT4 and primary hepatic stellate cells was increased, specifically towards FA-EVs. Prolonged incubation of TWNT4 cells with FA-EVs induced expression of proliferation markers and a myofibroblast-like phenotype. Though the expression of the collagen type 1 α1 gene did not change after FA-EV treatment, expression of the myofibroblast markers, e.g., α-smooth-muscle-cell actin and TIMP1, was significantly increased. We conclude that EVs from steatotic hepatocytes can influence the behavior, phenotypes and expression levels of remodeling markers of stellate cells and guides their directed migration. These findings imply EVs as operational, intercellular communicators in the pathophysiology of steatosis-associated liver fibrosis and might represent a novel diagnostic parameter and therapeutic target.

## 1. Introduction

Non-alcoholic fatty liver hepatitis (NASH), recently renamed metabolic associated fatty liver disease (MAFLD), is the leading cause of liver disease in the Western world [1,2]. Obesity, dyslipidemia and type 2 diabetes mellitus are the main pathogenic triggers [3]. MAFLD is clinically defined by atypical aspects during imaging and increases in transaminases and cholestasis parameters. It leads to liver fibrosis and long-term progression to cirrhosis [4,5], which is associated with numerous potentially lethal complications, such as gastrointestinal bleeding, coagulation disorders and ultimately, the development of hepatocellular carcinoma [6]. The emergence and progression of MAFLD at the cellular level is the subject of many current research projects. Although the diverse aspects of the pathophysiology of MAFLD remain to be completely characterized, it is increasingly clear that inflammation is a central element [7]. For example, altered microbiota and gut permeability during obesity may result in low-grade systemic inflammation contributing to the development of MAFLD [8,9,10]. In addition, dyslipidemia and adipocytokines from visceral adipose tissue are considered to initiate and propagate liver inflammation [4]. Subsequent drivers are fatty acid, cytokine and cholesterol-induced stress responses in Kupffer cells and hepatocytes, leading to the release of fibrogenic factors, e.g., transforming growth factor beta (TGF-β) [11], platelet-derived growth factor (PDGF) [12,13] and chemokines, such as CCL2 and CCL5. These factors consecutively promote activation and differentiation of hepatic stellate cells into myofibroblasts and the recruitment of additional inflammatory cells, e.g., classical monocytes into the liver [14,15,16,17,18]. Hepatic stellate cells are central in the pathophysiology of liver fibrosis by responding to the fibrogenic and morphogenic signals mainly derived from hepatocytes, infiltrated bone marrow-derived macrophages and resident Kupffer cells [5,19]. Transdifferentiation of stellate cells is associated with the expression of myofibroblast markers, e.g., α-smooth-muscle-cell actin (α-SMA), matrix metalloproteinases (MMPs), vimentin and glial fibrillary acidic protein (GFAP) and extracellular matrix components, such as collagen types I and III [20,21]. Excessive extracellular matrix production along with ineffective resorption are the key processes in organ fibrosis [22]. Extracellular vesicles (EVs) are small cell fragments measuring up to 1000 nm encased by fragments of the endosome or surface membranes of their parental cells [23,24]. EVs are released into the immediate environment of the parent cells and may be transported through the bloodstream to distal areas of the organism. Their contents are composed of proteins, lipids and nucleic acids of the parent cell. EVs can be incorporated by target cells and may thereby influence the behavior of the recipient cell. Thus, EVs can be considered as a physiological mechanism for cell–cell communication not requiring direct spatial interaction. Previous work by us and by others has demonstrated that EVs can alter the behavior of target cells [25,26]. Hepatocytes have been found to release increased amounts of EVs after treatment with fatty-acid-activating macrophages [27]. Moreover, the release of EVs by hepatocytes was implied to activate hepatic stellate cells and promote fibrosis in a mouse model of CCl4 liver injury [28]. In the present work, the influence of EVs derived from fatty hepatocytes on hepatic stellate cells is evaluated. Our hypothesis was that hepatocytes stressed by excess fatty acids can transmit signals through EVs to hepatic stellate cells, which may facilitate the transformation of resting stellate cells to active myofibroblasts and thus promote initiation and progression of the fibrogenic process during MAFLD.

## 2. Materials and Methods

### 2.1. Patient Samples, Reagents and Cell Lines

TWNT-4, immortalized human stellate cells, kindly provided by Professor Scott Friedman, Icahn School of Medicine at Mount Sinai, NY, USA [29], and HepG2, human hepatoma cells, were cultured in filtered Dulbecco’s modified eagle’s medium (DMEM) with 2% or 10% fetal bovine serum (Pan-Biotech GmbH, Aidenbach, Germany) and 1% penicillin and streptomycin (Thermo Fisher Scientific, Waltham, MA, USA) at passages 3–8 in 37 °C and 5% CO${}_{2}$ in a humidified incubator. For induction of a steatotic phenotype, the cells were treated with 10% linoleic–oleic acid (Sigma-Aldrich, St. Louis, MO, USA) for 24 h. Primary human stellate cells were obtained from Lonza (Basel, Switzerland) and cultured in human stellate cell growth medium (Lonza) according to the manufacturer’s instructions. Serum was taken from female and male patients diagnosed with MAFLD (*n* = 5, m:f 3:2, ages: 43–77 years, mean age 52.6 years) and controls (*n* = 5, m:f 3:2, ages 33–56 years, mean age 44.8 years) by venipuncture after informed consent was obtained. The study was approved by the University Hospital RWTH Aachen ethics board. Normal pooled plasma was obtained from healthy volunteers as described [30]. The antibodies used in this study are directed against human antigens and listed in Table 1.

### 2.2. Isolation of EVs

EVs were isolated from expired platelet packs, normal pooled plasma, or conditioned cell culture supernatants of up to 300 mL as previously described [26] according to ISEV recommendations [31,32]. The medium was filtered through 0.8 μm filters (Sartorius, Göttingen, Germany) by gravity flow and subsequently centrifuged at 20,000× *g* for 1 h at 16 °C. The pellet was resuspended in Hepes buffer pH 6.6 (10 mM Hepes, 136 mM NaCl, 2.7 mM KCl, 2 mM MgCl${}_{2}$, 5 mM glucose and 0.1% BSA).

### 2.3. Quantification of EVs

Concentrations and vesicle size distributions were determined by NTA (nanoparticle tracking analysis, Malvern NanoSight NS300, Malvern Technologies, Malvern, UK or ZetaView, Particle Metrix, Wildmoos, Germany) equipped with a 488 nm laser at 1:100 and 1:1000 dilutions. Every sample was counted 5 times for 1 min at 20 °C. The camera level was set at 16, and traces were analyzed using NTA 3.1.54 (Malvern NanoSight NS300, Malvern Technologies, Malvern, UK) or ZetaView 8.05.11 software (ZetaView, Particle Metrix, Wildmoos, Germany) with a detection threshold of 5, as described in [26]. To compare the cellular release of EVs between treatments, the absolute counts of EVs in the cell media samples were normalized to the cell counts in the respective culture vessels.

### 2.4. Cryo-Transmission Electron Microscopy (Cryo-TEM)

The preparations of platelet EVs were visualized by the cryo-TEM method. A thin aqueous film was formed by applying a 5 μL droplet of the suspension to a bare specimen grid. Glow-discharged holey carbon grids were used. After the application of the suspension, the grid was blotted against filter paper, leaving a thin sample film spanning the grid holes. These films were vitrified by plunging the grid into ethane, which was kept at its melting point by liquid nitrogen, using a Vitrobot (Thermo Fisher Scientific/FEI Company, Eindhoven, The Netherlands); the sample was kept before freezing at 95% humidity. The vitreous sample films were transferred to a microscope Tecnai T12 Spirit (FEI Company) using a Gatan cryotransfer. The images were taken at 200 kV with 53,000× magnification using a 4096 × 4096 pixel CCD Eagle camera (FEI Company) at a temperature between −170 and −175 °C, in low-dose imaging conditions.

### 2.5. Cell Migration in a Boyden Chamber

Home-made filter chambers were assembled with Whatman Nuclepore™ Track-Etched filters (Merck/Sigma Aldrich, St. Louis, MO, USA), coated with gelatine prior to use. The lower chamber was filled with DMEM/2% FBS, along with the chemoattractant or undiluted patient and control sera (800 μL). CCL5 at 10 ng/mL was used as a positive control for HSC chemotaxis (PeproTech, Hamburg, Germany). The upper chamber was filled with 2 × 10${}^{5}$ TWNT4 or primary human stellate cells in 250 μL DMEM/2% FBS. After 4 h at 37 °C and 5% CO${}_{2}$, membranes were stained using the hemacolor quick stain kit (Merck) and cells were quantified under a microscope. Five view fields were counted per filter. For every probe three filters were evaluated.

### 2.6. Qualitative and Quantitative DNA Analysis by qRT-PCR

For RNA isolation, peqGOLD TriFast™ was used according the manufacturer’s instructions (VWR international, Darmstadt, Germany). RNA-containing pellets were resuspended in nuclease-free water, and concentrations were determined using a NanoDrop spectrometer (Thermo Fisher). cDNAs were generated using a Maxima First Strand cDNA Synthesis Kit (Thermo Fisher) as described in [33]. Quantitative qRT-PCR analysis was performed using GoTaq^®^, qPCR Mastermix (Promega, Madison, WI, USA) with primers for PLIN2, KI67, PCNA, ASMA, COL1A1, TIMP1, TGFB1, 18S and MMP2. The cycler program was 2 min at 50 °C, 10 min at 95 °C and 1 min at 60 °C for 40 cycles using a qPCR system: Applied Biosystems 7300 (Thermo Fisher). Delta-deltaCT values were calculated relative to the housekeeping gene 18S, and results were normalized to controls.

### 2.7. Protein Concentration Determination

Protein concentrations were determined using the BCA method (Bio-Rad, Hercules, CA, USA). Cells were cultured in 6-well plates, washed 2 times with PBS, lysed in 300 μL RIPA-buffer and incubated with reagents. Then, absorbance was measured at 560 nm by a microplate reader. Concentrations were calculated using a standard curve prepared with BSA. Concentrations of CCL5 in EV were determined in the presence of 0.1% tween-20 by an enzyme-linked immunosorbent assay (ELISA) according to the manufacturer’s instructions (R&D Systems, Minneapolis, MN, USA).

### 2.8. SDS-PAGE and Western Blot

For protein analysis, NuPAGE 4–12% Bis-Tris Gel SDS-PAGE was performed in MOPS-SDS running buffer (Thermo Fisher) for 60 min at 160 V, and proteins were subsequently transferred to nitrocellulose (Whatman, Cytiva, Marlborough, MA, USA) for 1 h at 100 V. The membranes were blocked using 5% BSA and incubated with primary and peroxidase-conjugated secondary antibodies prior to detection using enhanced chemiluminescence.

### 2.9. Cell Viability

TWNT4 cells were added at 50,000 per well in a 96-well microplate and cultured overnight under the conditions described above. Media were replaced with fresh medium containing EVs at 300 per cell or CCL5 (25 ng) and cultured for a further 12 h. As a positive control, cells were treated with 30% H${}_{2}$O${}_{2}$ to induce cell death. Cell-Titer-Blue-reagent (Promega, Madison, WI, USA) was added for 2 h, followed by measurement of the fluorescent signal at 560 em/590 ex nm.

### 2.10. Oil-Red-O and Phalloidin Staining

Cells were seeded in 4-well chamber slides at 50,000 cells per well. After 24 h, cells were stimulated overnight with EVs (300 per cell) and CCL5 (10 ng/mL). To investigate lipid uptake, linoleic–oleic acid (300 μM) was added to HepG2 cells, and culturing was continued for a further 24 h. Medium was removed. Cells were washed with PBS, fixed with 4% formaldehyde for 5 min and washed again. Intracellular lipid vacuoles were stained in filtered 0.12% Oil-Red-O in 20% isopropanol for 1 h. Cells were washed with water. After staining of nuclei for 25 s in Mayer’s hematoxylin solution (Sigma Aldrich), cells were washed again and covered with a cover slip treated with glycerine–gelatine. Evaluation of Oil-Red-O uptake was performed by light microscopy. To visualize the cytoskeleton in TWNT4 cells, medium was removed, and cells were washed with PBS, fixed in 4% formaldehyde and washed again with PBS prior to permeabilization by 0.1% Triton-X100 in PBS for 15 min. After washing with PBS, cells were blocked in 1% BSA-PBS for 1 h. Anti-collagen antibody (Sanbio BV, Uden, The Netherlands) was given to the cells for one hour. Fluorescein-conjugated phalloidin (1U in 1% BSA/PBS) was added for 30 min. The cells were washed with PBS and nuclei were visualized by DAPI reagent. Evaluation of cytoskeletal staining was performed by fluorescence microscopy at 490/530 nm. Corrected total cell fluorescence (CTCF) was calculated using Image J 1.5 [34] as described in [35].

### 2.11. Cell Proliferation Measurement

The 5-bromo-2${}^{\prime }$-deoxyuridine (BrdU) assay (Thermo Fisher) was used for quantification of proliferating cells according to the manufacturer’s instructions.

### 2.12. Flow Cytometry Analysis of CCR5 Surface Expression

TWNT-4 cells were seeded at 100,000 cells per well in a 96-well plate and were allowed to rest overnight. Cells were treated with 10 ng/mL CCL5 or 300 EV/cell for 24 h. Cells were detached, filtered using a cell strainer, washed in PBS and stained in the dark with rat anti-human/mouse CCR5 FITC-conjugated antibody (clone HEK/1/85a from Thermo Fisher) for 1 h at 4 °C. The cells were subsequently stained with Fixable Viability Dye eFluor450 (eBioscience) for a further 30 min. A negative control only contained viability dye. Cells were gated according to FSC and SSC parameter and live cells using a FACSCanto II (Becton Dickinson, Franklin Lakes, NJ, USA). Data were processed and analyzed using FlowJo 7.6 software (Becton Dickinson).

### 2.13. Statistics

All authors had access to the study data and have reviewed and approved the final manuscript. All data generated or analyzed during this study are included in this article. Experiments were performed at least 3 times independently, and there was a minimum of three technical replicates. Experimental data are represented as mean ± SEM. Statistical analysis of the data was performed by two-tailed *t*-test or ANOVA with Bonferroni post hoc test, as indicated in the figure legends. A *p*-value below 0.05 was considered significant. Statistical analysis was performed with Graphpad Prism 9 software (San Diego, CA, USA)

## 3. Results

### 3.1. HepG2 Cells Release Increased Amounts of EVs under Steatogenic Conditions

Cells of the human hepatocellular carcinoma cell line HepG2 were treated with a mixture of linoleic and oleic acids (fatty acids, FA). The release of EVs by HepG2 cells under baseline conditions (no treatment; C-EVs) and after treatment with FA (FA-EVs) was assessed in conditioned culture media collected over 24 h. Characterization of the EVs derived from those HepG2 cells by NTA showed an average size of 0.17 μm for both conditions (Figure 1A,C), which was also supported by cryo-TEM, revealing spherical, membrane enclosed, mainly unilamellar EVs (Figure 1B,D). The EV were morphologically comparable between EVs isolated after either treatment. Western blot analysis revealed the presence of exosome markers Alix, TSG101 and synthenin [36] in C-EVs and FA-EVs, similarly to the case of control EVs isolated from platelets or normal pooled plasma (Figure 1E). This suggests that the EV preparations contained both larger microvesicles and exosomes. The HepG2 cells were found to constitutively release extracellular vesicles (EVs) in the culture media (Figure 1F). Interestingly, and in line with previous studies [27], the number of EV released per cell was significantly elevated after treatment with FA (Figure 1E). The steatotic phenotype was confirmed by enhanced staining with Oil-Red-O and increased cellular triglyceride content (Appendix B Figure A1A). In addition, the expression levels of the lipid droplet-associated perilipin (*PLIN-1*) and fatty acid binding protein 1 (*FABP1*) were increased (Appendix B Figure A1B–D).

### 3.2. EVs from HepG2 Cells Modulate Migration of Stellate Cells

Steatotic hepatocytes have been shown to secrete several factors, including distinct cytokines and EVs, which could be measured in the sera of individuals with fatty liver disease [27,37]. These factors might directly impact biological features of hepatic stellate cells, such as their migration to sites of release, and thereby modulate the distribution of hepatic stellate cells and fibrotic response in the tissue. In a Boyden chamber setup, TWNT-4 cells—TWNT-4 is a human hepatic stellate cell line—and primary human hepatic stellate cells (HSCs) were allowed to migrate towards the established chemoattractant CCL5 in the sera of healthy individuals and those with clinically diagnosed MAFLD. Interestingly, the sera from healthy individuals did not induce migration of TWNT-4 cells, whereas TWNT-4 cells showed pronounced migration towards sera from MAFLD patients (Figure 2A). However, of course, a myriad of factors within the sera might contribute to the enhanced migration of hepatic stellate cells to some extent.

To specifically address the impact of HepG2-derived EVs on hepatic stellate cell function, we went back to the controlled in vitro setting and assessed EV-associated modulation of distinct hallmarks of hepatic stellate cell biology. We started with analyzing C-EV and FA-EV-induced TWNT-4 migration. Stellate cells are activated by and migrate to various molecular cues. For example, the chemokine CCL5 was found to trigger migration of TWNT-4 stellate cells and to be involved in the development of experimental liver fibrosis [17]. In line with the findings in the patient serum setting, FA-EVs triggered chemotactic migration of TWNT-4 cells to the same extent as CCL5, whereas C-EVs did not trigger chemotaxis (Figure 2B). Therefore, further analyses were mainly focused on FA-EV. Interestingly, when the TWNT-4 were co-cultured with FA-EVs prior to chemotaxis, the cells no longer migrated towards CCL5 (Figure 2C). To rule out that the impaired migration was due to the loss of expression of the CCL5 receptor CCR5 on the surface of the cells, we performed flow cytometry to analyze the surface expression levels of CCR5 on the TWNT-4 cells, which remained unaltered after co-culturing with EVs of either origin (Appendix B Figure A1E,F). Of note, primary HSCs were also found to migrate towards FA-EVs to a similar extent as they did towards CCL5, further supporting the chemoattractant potential of steatotic HepG2-derived FA-EVs (Figure 2D). However, ELISA measurements revealed that the CCL5 levels in both C-EV and FA-EV were below the detection limits (not shown).

### 3.3. EVs from HepG2 Cells Increase the Proliferation of TWNT-4 Stellate Cells

Activation of hepatic stellate cells leads to mitogenic responses and increased proliferation [4,5]. We therefore next assessed the impact of HepG2-derived EVs on hepatic stellate cells’ proliferation. TWNT-4 cells were stimulated with CCL5 and with FA-EVs for 24 h, and proliferation was determined by BrdU incorporation and by the mRNA expression of the proliferation markers *Ki-67* and *PCNA*. Expression levels of *Ki67* and *PCNA* were increased in TWNT-4 cells after treatment with FA-EVs. The BrdU incorporation was also significantly enhanced (Figure 3A–C). The viability of the TWNT-4 cells was not reduced after treatment with CCL5 or FA-EV, although the combination of CCL5 and FA-EV led to a notable reduction in cell viability (Figure 3D). In contrast to the TWNT-4 cell line, the treatment of primary HSCs with FA-EVs did not increase the expression of the proliferation marker *PCNA* (Figure 3E).

### 3.4. EVs from Steatotic HepG2 Modulate the Expression of Matrix Myofibroblast Markers and Remodeling Markers in Stellate Cells

Stellate cells are crucially involved in fibrotic liver remodeling and are considered as the central effectors of liver fibrosis [4,5]. Once hepatic stellates are activated, they undergo a process of transdifferentiation and gain a myofibroblast-like phenotype associated with the synthesis and release of collagen and matrix remodeling factors such as TIMP-1. Since hepatic stellate cells are in direct proximity to hepatocytes, possible influences of EVs derived from HepG2 cells under steatotic conditions on transdifferentiation and collagen production were investigated in TWNT4 cells. After treatment of TWNT4 cells with CCL5 and FA-EVs for 24 h, the fraction of elongated cells that had assumed a myofibroblast-like morphology significantly increased by almost 1.5-fold (Figure 4A–D). In addition, expression levels of myofibroblast markers such as α-*SMA*, GFAP and vimentin were significantly increased after treatment with FA-EVs (Figure 4E–G). Increased expression of α-SMA was also observed in primary HSCs treated with FA-EVs (Figure 4H).

In contrast, collagen production visualized by fluorescence microscopy and mRNA expression levels of the collagen type 1 α1 gene (*COL1A1*) and *TGF*-β did not significantly change after FA-EV treatment of TWNT4 cells (Figure 5A–E). In addition, the expression of collagen tended to decrease after treatment of primary HSCs with FA-EVs (Figure 5I). However, expression of the matrix remodeling markers *TIMP1* and *MMP2* increased significantly after incubation of TWNT4 cells with FA-EVs (Figure 5F–H).

## 4. Discussion

In this study, the effects of EVs harvested from FA-treated HepG2 cells (FA-EVs) on stellate TWNT4 cells were investigated. In accordance with previous observations, treatment of HepG2 cells with lipotoxic compounds, e.g., lysophosphatidylcholine (LPC), or its precursors palmitic and oleic acids, leads to an increase in EV formation [27,37,38,39]. Given the important function of hepatic stellate cells in liver inflammation and the progression from steatosis to fibrosis, human TWNT4 cells and primary human HSCs were used as model systems to investigate the mechanisms behind the transition of hepatic steatosis to fibrosis.

In chemotaxis experiments, TWNT4 cells showed migration solely towards sera of individuals with MAFLD, whereas control sera did not induce any effects. Thus, MAFLD sera may contain increased amounts of inflammatory factors that induce responses in target cells, e.g., hepatic stellate cells. As previous studies have identified increased amounts of cytokines [27] or EVs [37] in sera from MAFLD patients, it was hypothesized that EVs derived from steatotic HepG2 cells might induce these effects in TWNT4 cells. Interestingly, both TWNT4 and primary HSCs were found to migrate only towards FA-EVs and CCL5 and not towards control EVs from HepG2 cells. A possible explanation is that the lipotoxic effects of the FA treatment result in altered packaging of the EV cargo, with an enrichment of components involved in inflammation or cell proliferation. For example, Hirsova and colleagues observed enrichment of the cell death- and inflammation-related tumor necrosis factor related apoptosis inducing ligand (TRAIL) in EVs isolated from lysophosphatidyl choline (LPC)-treated hepatocellular carcinoma Huh7 cells [27]. In other studies, it was observed that EVs from LPC-treated Huh7 cells were enriched in integrin β1 [37] or CXCL10 [39]. In a mouse model of liver fibrosis, exosomes were shown to activate toll-like receptor 3 in HSCs [28]. In addition to proteins, bioactive lipids, e.g., ceramides, might be enriched in EVs released from hepatocytes after treatment with fatty acids [38], and recent studies have also provided evidence of the transfer of microRNAs, carried by hepatocyte-derived exosomes to stellate cells [40,41]. Interestingly, although TWNT4 cells actively migrated towards a gradient of FA-EVs, culturing of TWNT4 with FA-EVs prior to the chemotaxis experiment abolished the migration of these cells towards CCL5, an established chemoattractant for these cells [17]. This was not due to alterations in the surface expression of CCR5, which was found to be equal on the cells of all treatments. It might be surprising that the FA-EVs induce chemotaxis in the short term and inhibit this process in the long term. However, once the stellate cells are at the site of inflammation, they may be kept in place by migration-inhibiting signals contained in the FA-EVs. In addition to local effects of fatty acid-induced EV release, EVs from damaged hepatocytes might also attract the stellate cells over a longer distance to the center of disease. It is tempting to speculate that this might be among the cause of the typical histological fibrosis in MAFLD cirrhosis, which presents as sinusoidal fibrosis. Thus, a chemotactic effect of the EVs, as suggested by our experiments, would cause the stellate cells to migrate along the liver sinusoids (Figure 6). The EVs are incorporated by the stellate cells once in contact at higher concentrations, which might lead to arrest of migration and production of extracellular matrix, causing the specific situation of collagen fibers along the sinusoids lined by steatotic hepatocytes—often termed “chicken wire” fibrosis [42].

Once hepatic stellate cells have arrived at the site of inflammation, proliferation and production of extracellular matrix increase, finally inducing the progression to liver fibrosis. Treatment of TWNT4 cells with FA-EVs led to an increase in BrdU incorporation and increases proliferation markers. Cell viability was not significantly affected by FA-EVs, although the combination of CCL5 and FA-EVs led to a notable reduction in cell viability. Interestingly, whereas the production of collagen by FA-EV-treated TWNT4 and primary HSCs did not significantly increase, the cells appeared to change phenotype to an elongated appearance typical of myofibroblasts [20,43]. In addition, the expression levels of several markers of a myofibroblast phenotype were found to be increased in TWNT4 and primary HSCs after FA-EV treatment. Our findings indicate that matrix production is not strongly affected by extracellular vesicles, but rather proliferation and differentiation. Taken together, FA-EVs have the potential to modulate the responses of recipient cells, such as HSCs, and these observations are in line with recent studies [40,41], supporting the concept of EV as mediators of cell-to-cell communication [24,44].

The question about the exact mechanism of how the EVs transfer their signals to recipient cells in this study remains open and difficult to elucidate, since the observed effects might be due to the concerted actions of several classes of bioactive compounds present in EVs (e.g., lipids, proteins and nucleic acids). A number of effectors carried by EVs were suggested to exert biological effects in the literature. An aforementioned study highlighted TRAIL as responsible for the pro-inflammatory and pro-fibrotic effects during fatty liver disease [27]. Others have identified particular micro-RNAs as mediators of liver tissue remodeling [45]. In our previous study focusing on EVs from platelets, over 500 proteins were identified using mass spectrometry, and the pro-inflammatory effects of platelet-EVs on smooth muscle cells could, at least partly, be narrowed down to the actions of chemokine CXCL4 [26]. Others have shown that bioactive lipids, e.g., LPC, also mediate the proinflammatory effects of EVs [46]. We initially hypothesized that CCL5 would be responsible for the differentiation and increased proliferation of the HepG2-derived EVs on cultured stellate cells. However, ELISA measurements showed undetectable CCL5 levels even in dense EV suspensions (10${}^{10}$/mL). Thus, the further characterization of the molecular effectors carried by EVs will be subject of future studies and might require lipidomics, proteomics and next generation sequencing in combination with contemporary bioinformatics.

Two recent studies implementing next generation sequencing have yielded novel mechanistic insights into the cellular mechanisms governing the development of MAFLD and liver fibrosis ([47,48], respectively). Interestingly, unsupervised clustering of single-cell sequencing data revealed the involvement of a variety of non-parenchymal cell types and their subpopulations. This analysis, for example, showed surprising involvement by endothelial cells, which exist in distinct subpopulations in MAFLD and fibrotic livers and might control liver metabolism and populate the fibrotic niche. In addition, a special type of macrophages expressing triggering receptors expressed on myeloid cells 2 (TREM2) was identified. Although no study that focused on the role of EVs in the development of this macrophage subset has been published so far, a role for EV-mediated signaling in this process can be envisioned. The study investigating cell populations in human and mouse MAFLD has taken efforts to unravel intercellular signaling between the identified cell types [47]. Here, hepatic stellate cells were found to be an important cellular hub, both in receiving and transmitting signals from and to hepatocytes, macrophages and endothelial cells. Although a role of EVs in the trafficking of these cellular signals has not been explicitly taken into account, the numerous recent studies that highlighted EVs as mediators make a strong case for their integral role in intercellular signaling leading to MAFLD and liver fibrosis.

In this study, EV-mediated signal transmission from FA-treated hepatocytes to hepatic stellate cells was investigated. Although other cell types are instrumental in the development of liver disease, hepatic stellate cells are central in the orchestration of the pathological responses of non-parenchymal cells in the liver. An excessive flow of hepatocyte-derived EV to macrophages and stellate cells might facilitate the development of MAFLD and liver fibrosis through their potential to modulate cellular phenotype and behavior. Thus, future investigation of this process might not only lead to potentially novel options for therapeutic intervention, but also to improved diagnostic insights through analysis of plasma EV numbers and content.

## Figures and Tables

**Figure 1 biomolecules-12-00698-f001:**
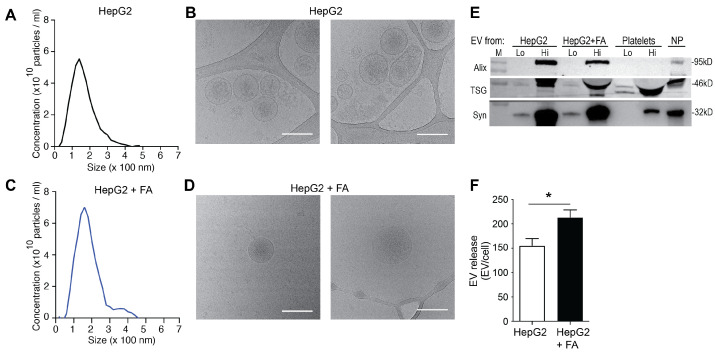
Treatment with fatty acids induces steatosis in HepG2 cells. NTA analysis (**A**,**C**) and representative electron micrographs ((**B**,**D**) at 53,000× magnification) of EVs in media of HepG2 cells collected under resting conditions (**A**,**B**) or after treatment with fatty acids (FA) for 24 h (**C**,**D**). Scale bars: 200 nm. (**E**) Western blot of the EV markers Alix, TSG101 and Syntenin in EVs isolated from HepG2 cells with or without FA treatment and from platelets (Lo: 10${}^{7}$, Hi: 10${}^{8}$ EV) or normal pooled plasma (NP, 6 × 10${}^{7}$ EV). (**F**) Expression of EV release as number of EVs per cell. * *p* < 0.05 with 2-tailed *t*-test, mean ± SEM (*n* = 3).

**Figure 2 biomolecules-12-00698-f002:**
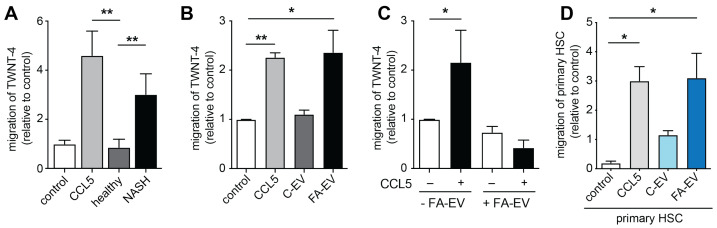
Migration of TWNT4 and primary stellate cells towards patient sera and EVs released from HepG2 cells. Chemotaxis of TWNT4 cells towards CCL5 or (**A**) sera from individuals with or without MAFLD, or (**B**) towards EVs harvested from HepG2 under resting conditions (C-EVs) or after fatty acid treatment for 24 h (FA-EVs). (**C**) Chemotaxis of TWNT4 cells towards CCL5 with or without pre-treatment with FA-EVs for 24 h. (**D**) Chemotaxis of primary human hepatic stellate cells (HSCs) towards C-EVs or FA-EVs (light and dark blue bars, respectively). Five view fields were counted per filter and normalized to the control. * p<0.05, ** p<0.01 with one-way ANOVA and Bonferroni post hoc test; mean ± SEM (*n* = 3).

**Figure 3 biomolecules-12-00698-f003:**
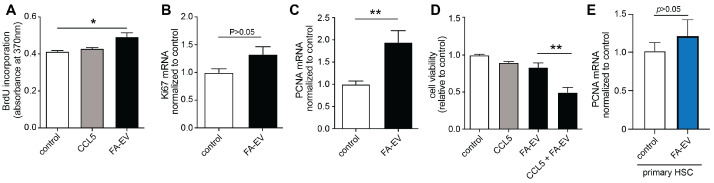
Proliferation and cell viability of TWNT4 cells and primary HSC after EV treatment. Proliferation of TWNT4 cells measured by BrdU incorporation after treatment with CCL5 or FA-EVs (**A**). Expression of the proliferation markers *Ki67* (**B**) and *PCNA* (**C**) after treatment of TWNT4 with FA-EV. (**D**) Viability of TWNT4 cells after treatment with CCL5, FA-EVs, or a combination of CCL5 and FA-EVs. (**E**) Primary HSCs were treated with or without FA-EVs, and mRNA expression levels of PCNA were measured. * p<0.05, ** p<0.01 with post hoc ANOVA and Bonferroni test or 2-tailed *t*-test; mean ± SEM (*n* = 3–4).

**Figure 4 biomolecules-12-00698-f004:**
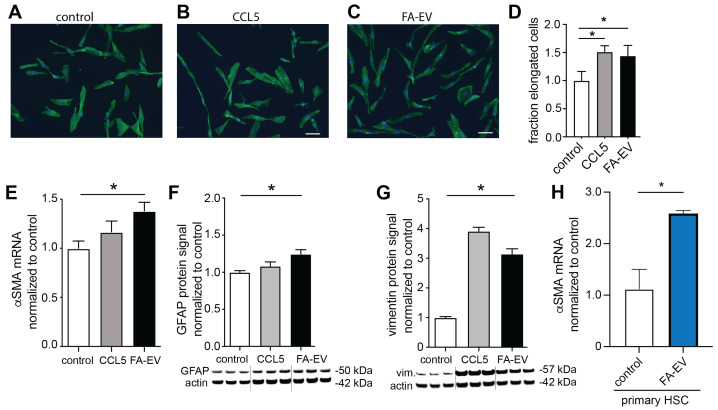
Morphology and phenotypic markers of FA-EV–treated TWNT4 cells and primary HSC. Representative phalloidin staining after treatment of TWNT4 cells with vehicle (**A**), CCL5 (**B**) or FA-EVs (**C**). Scale bar: 100 μm. (**D**) Quantitation of elongated cells from the micrographs (*n* = 3). mRNA expression of α-*SMA* (**E**), and antigen levels of GFAP (**F**) and vimentin (**G**) after treatment with vehicle, CCL5 or FA-EVs. (**H**) Primary HSCs were treated with or without FA-EVs, and mRNA expression levels of α-*SMA* were measured. * p<0.05 with ANOVA and Bonferroni test; mean ± SEM (*n* = 3–5).

**Figure 5 biomolecules-12-00698-f005:**
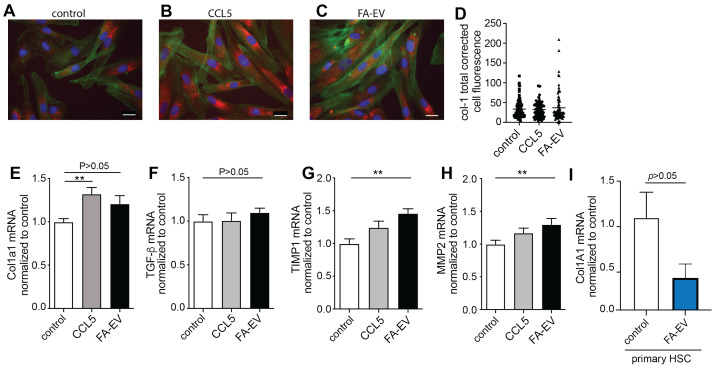
Expression of fibrotic markers in FA-EV–treated TWNT4 cells and primary HSC. Representative phalloidin staining and collagen staining after treatment of TWNT4 cells with vehicle (**A**), CCL5 (**B**) or FA-EV (**C**); scale bar: 20 μm. (**D**) Quantitation of red (collagen) fluorescence. mRNA expression levels of *COL1A1* (**E**), *TGF*-β (**F**), *TIMP1* (**G**) and *MMP2* (**H**), after treatment with vehicle, CCL5 or FA-EV. (**I**) Primary HSCs were treated with or without FA-EVs, and mRNA expression levels of COL1A1 were measured. ** p<0.01 with ANOVA and Bonferroni test; mean ± S EM (*n* = 3–5).

**Figure 6 biomolecules-12-00698-f006:**
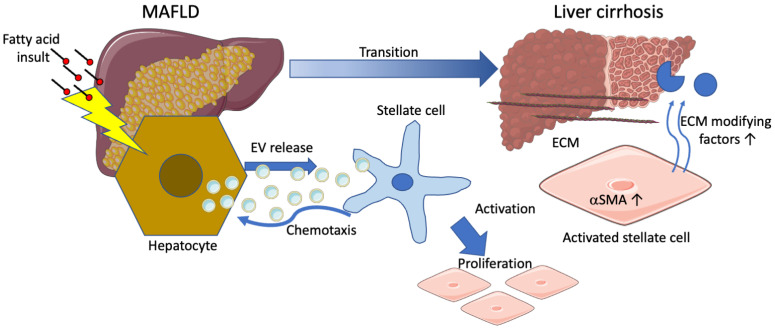
Schematic overview of the effects of EVs on HSCs during liver fibrosis pathogenesis. During MAFLD, a chronic insult of fatty acids may lead to increased EV release by hepatocytes, which causes chemotaxis of HSCs along the liver sinusoids and activation and proliferation of HSCs, leading to a myofibroblast-like phenotype with increased α-SMA, a marker, and increased release of extracellular matrix (ECM)-modifying factors.

**Table 1 biomolecules-12-00698-t001:** Antibodies, species and supplier information.

Antigen	Species, Type	Supplier
Alix	mouse monoclonal	Bio-Rad, OTI1A4
β-actin	rabbit monoclonal	Cell Signal. Tech., D6A8
CCR5	rat monoclonal	ThermoFisher, HEK/1/85a
Collagen I-V	rabbit polyclonal	Sanbio BV, PS046
GAPDH	mouse monoclonal	Bio-Rad, 6C5
GFAP	rabbit polyclonal	Abcam, ab7260
PLIN1	rabbit monoclonal	Abcam, EPR3753(2)
α-SMA	rabbit monoclonal	Cell Signal. Tech., D4K9N
Synthenin	rabbit monoclonal	Abcam, EPR8102
TSG101	rabbit polyclonal	Sigma, T5701
Vimentin	rabbit monoclonal	Abcam, EPR3776
Anti-mouse-peroxidase	goat polyclonal	Agilent, P044701-2
Anti-rabbit-peroxidase	goat polyclonal	Agilent, P044801-2

## Data Availability

The source data of this study can be obtained from the corresponding authors on reasonable request. Representative images are supplied in a Appendix A.

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
