# Peer review of "Extracellular Vesicles from Steatotic Hepatocytes Provoke Pro-Fibrotic Responses in Cultured Stellate Cells"

_biomolecules, 2022, doi:10.3390/biom12050698_

Round 1

Reviewer 1 Report

This study showed that extracellular vesicles from steatotic hepatocytes provoke pro-fibrotic responses in hepatic stellate cells (HSCs). This manuscript was interesting, well-written and used appropriate methods. I have minor questions before publication.

  1. Author used primary HSCs only for migration, not for evaluating pro-fibrotic responses. Is there any reason for it?
  2. I suggest analyzing the contents in extracellular vesicles to figure out the causative growth factors or cytokines which can increase the pro-fibrotic responses.

Author Response

Please see attached file with our response to reviewer 1.

Reviewer 2 Report

This is an interesting paper tried to address pro-fibrotic response of EV from steatotic hepatocytes on  cultured stellate cells. Overall, it is an imprtant topic for the field if the proper mechanistic approach is provided:

1- The EV induced phenotype change was described well but there is no mechanism on pro-fibrotic effect. Why EV induce profibrotic effect? how? Is it related on the secretome of the cells after EV treatment? This is something that the author needs to address it. 

2- Detection of pro-fibrotic effect was in-complete. The respected author should determine the secreted collagen and fibronectin using WB. 

3- The mechanism of proliferation should be determined. 

Technical Correction:

1- The cell number in 96 well plate is too high and could potentially induce significant error in MTT assay.

2- All information regarding the antibodies should be provided. 

3- n=3 of whole data should be provided as supplementary figures. 

Author Response

Please see attached file with our response to reviewer 2.

Round 2

Reviewer 2 Report

Although the respected authors were not able to answer some of my suggestions (1, 3 comments) with experimental approach, their answers and changes of the tone in the manuscript were very good. The current paper will open several opportunities for future investigations as explained by respected authors. 

Author Response

Thanks.